# ACTIVATIONS ARE NOT CHEAP FOR LoRA, WEIGHTS ARE

## ABSTRACT

LoRA has become the prevailing technique for finetuning large neural networks with limited computational resources. Historically, activations have been regarded as small and computationally inexpensive to manipulate—a view reflected by LoRA, which leverages this assumption and adds a low-rank term to intermediate activations. However, in the era of modern large language models (LLMs) and diffusion models, this notion has been challenged by the desire for increasing context lengths and smaller models, a trend which inevitably leads activations to consume more memory than the model weights themselves. Surprisingly, when finetuning a 1B model with a context length greater than 2048, we find that LoRA finetuning uses more memory than full-parameter finetuning. This study demonstrates that manipulating additional model weight representations within the computation graph in parameter-efficient finetuning techniques can often be more memory-efficient than operating on the activations. We provide a semantically-equivalent computation graph reformulation for LoRA, and other popular PeFT techniques, which saves memory and trains faster, advancing the Pareto-frontier for finetuning tasks that can be achieved on consumer hardware. Under practical conditions, this reformulation provides up to a **1.4x reduction in max memory usage and latency** for LoRA finetuning across various language and diffusion transformers without affecting the predictive performance of the technique.

## 1 INTRODUCTION

The release of the transformer architecture (Vaswani et al., 2017) inspired the widespread scaling of language models with billions or trillions of parameters. This unprecedented scale ushered in a fundamental paradigm shift in language modeling: it was no longer necessary to train models from scratch when generalist pre-trained models could be finetuned for higher performance on downstream tasks. It is no coincidence that as these models continued to scale, the field has taken a significant effort to understand how to finetune them efficiently, as doing so otherwise often meant consuming hundreds of gigabytes of GPU memory.

The existence of pretrained LLMs led to the creation of LoRA (Hu et al., 2021), a finetuning technique that focuses not on optimizing the parameters of the target model but instead on a small set of new parameters that can be merged into the original model without extra cost during inference. LoRA has now become a bedrock technique for efficiently finetuning LLMs. It introduces a pair of low-rank matrices that are forward-passed in parallel to a given model layer; these activations are summed and passed forward to the rest of the network. This approach can offer a significant memory saving over full-parameter finetuning, in which all original network parameters are optimized, as well as regularizes the model to forget less pre-training information (Biderman et al., 2024). LoRA achieves memory savings by eliminating the need to store the optimizer's state, including the first-order and second-order moments required by optimizers like ADAM. LoRA achieves 3x memory saving when fine-tuned on GPT-3 with 175B parameters (Hu et al., 2021).

The field of language modeling has seen substantial growth in recent years. New architectures, larger corpuses of multimodal data, and increased FLOP scaling have led to smarter, more capable models. Irrespective of these trends, the use of LoRA has largely persisted, and it continues to be a popular technique for finetuning the latest available models.

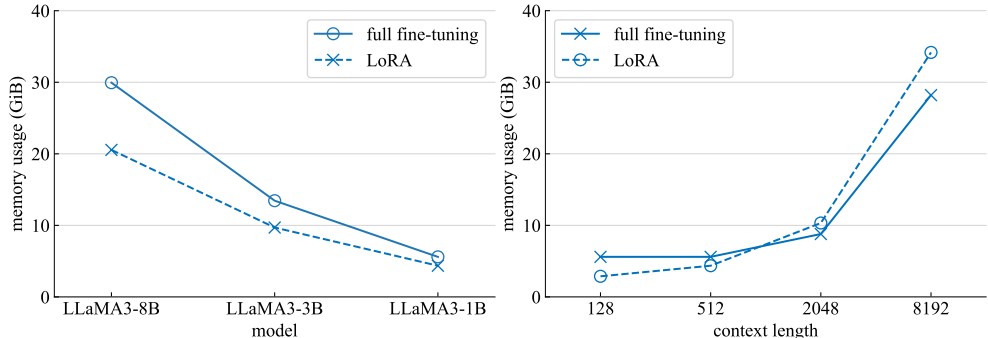

Figure 1: **LoRA saves less GPU memory.** In comparison to full-parameter finetuning, LoRA exhibits less memory savings on smaller models. Furthermore, LoRA incurs higher memory consumption than full finetuning when applied to longer context lengths.

However, the memory saving of LoRA might not persist as there are two emergent trends in the development of LLMs: *smaller models* and *longer context length.* These two trends are characterized by the creation of LLaMA-3.2-1B (Dubey et al., 2024), which is, at the time of writing, the smallest model in LLaMA series and supports a 128K context length. As LoRA's memory savings come from not storing weight-sized tensors in optimizer states, a smaller model would necessarily mean less memory saving. Furthermore, since LoRA induces additional activations, when activations are large due to long context length, this overhead may dominate the memory saving of LoRA. As shown in figure 1, LoRA saves less (or even increases) the memory usage compared to full finetuning when applied on smaller models.

One may ask the question: why use LoRA in these scenarios at all when full-parameter finetuning is possible? Critically, LoRA acts as a regularizer – it mitigates forgetting (Biderman et al., 2024). Where low-memory devices are used, compression techniques such as $NF4$ (Dettmers et al., 2024) can make the original weights non-differentiable, preventing full-parameter finetuning. Additionally, as we show is the case in Table 5 for diffusion models, full-parameter finetuning can have larger latency than LoRA. Also, LoRA adapters are lightweight and preferable when it is necessary to quickly swap them in and out of memory.

While larger models continue to scale, efforts have been made to develop small models based on large models. For example, Phi (Gunasekar et al., 2023), a lightweight model, is trained using synthetic data from GPT-3.5, and LLaMA-3.2-1B is developed using both pruning and distillation from larger LLaMA variants. Methods are developed to efficiently use both weights and outputs of pretrained large models to develop smaller models (Xia et al., 2024; Xu et al., 2024). In fact, context length supported by pretrained LLMs has indeed grown larger over years as shown in figure 2.

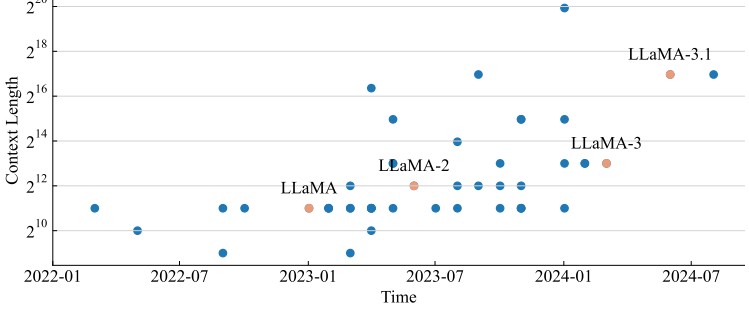

Figure 2: **Longer context lengths.** As time has progressed, the maximum context lengths, in terms of tokens, has trended upwards. This trend has serious implications for the amount of memory it takes to finetune transformers moving forward.

With these trends in mind, techniques like LoRA and its variants lose their memory efficiency as they produce and store even larger intermediate activations for backpropagation. It may soon be-

come evident that activation-based parameter efficient finetuning approaches, once champions for accessibility, might be too memory-intense for current and future models.

Will LoRA and other activation-based approaches still be usable at the current trend rate? In this study, we establish a semantically equivalent computation graph for LoRA and other PEFT techniques that depends less on these trends and drastically decreases their memory and latency overhead without affecting performance or changing their meaning. We begin by establishing a memory complexity analysis of activations versus weights, which leads us to a key observation that modifying weight representations requires less memory than modifying activations.

Leveraging this insight, we reformulate these techniques using simple, few-line changes to their definitions. We test our weight-based reformulation across a suite of popular modern LLMs, such as LLaMA 3.2, as well as diffusion transformers (Peebles & Xie, 2023), and observe a 1.4x reduction in memory and latency.

This reformulation is widely applicable and effective at saving memory and reducing latency. Its implementation pushes forward the Pareto-frontier of what can be achieved on consumer GPUs without sacrificing accuracy. We hope that these insights can offer a new theoretical perspective on designing future parameter-efficient finetuning techniques while practically enabling broader access to LLM finetuning.

## 2 MOTIVATION

### 2.1 PRELIMINARIES

**Low-rank adaptation (LoRA).** LoRA (Hu et al., 2021) inserts two trainable low-rank matrices $A \in \mathbb{R}^{m \times r}$ and $B \in \mathbb{R}^{r \times n}$ for every targeted weight $W \in \mathbb{R}^{m \times n}$ in a pretrained model. During a forward pass of the model, they are used to construct an additional projection that is added to the hidden state in the form:

$$y = Wx + BAx$$

where $BAx$ term would be scaled by a constant factor $c$ before added to $Wx$. During finetuning, only the parameters $A$ and $B$ are updated, while all weights from the pretrained model are frozen. During inference, $A$ and $B$'s product is merged back to $W$, and thus does not increase the inference latency.

**(IA)$^3$.** (IA)$^3$ (Liu et al., 2022) is a PEFT method that learns rescaling factors for activations in the network. Given a frozen weight $W \in \mathbb{R}^{m \times n}$, bias $b \in \mathbb{R}^{m \times n}$, input $x \in \mathbb{R}^{n \times 1}$, (IA)$^3$ would insert a trainable vector $l \in \mathbb{R}^{m \times 1}$ to directly rescale the activations:

$$y = (Wx + b) \odot l$$

where $\odot$ means the hadamard product which performs column-wise scaling.

### 2.2 ACTIVATIONS

Activations are the tensors passed forward between layers in a neural network. They are retained during training for backpropagation. In a transformer-based (Vaswani et al., 2017) model, an intermediate activation is typically a three-dimensional tensor of shape (B, S, D), where $B$ is *batch size*, $S$ is *sequence length*, and D is *hidden size*.

$$\text{Size of Activation} = B \times S \times D \tag{1}$$

$D$ would remain fixed for the same pretrained model. In contrast, *batch size* and *sequence length* could differ for different training runs.

**Batch size.** Larger batch size produces comparatively more reliable gradient direction, providing a more accurate approximation for the whole-dataset gradient (Liu et al., 2023). The linear-scaling rule (Goyal et al., 2017) between batch size and learning rate also suggests that larger batch size allows for larger steps in training and stabilizes training. In the context of LLM training and finetuning, a common practice is to use the maximum batch size that GPU memory allows.

**Sequence length.** Pretrained LLM's context length grows over time. For example, LLaMA (Touvron et al., 2023a) is pretrained using a context length of 2048, and LLaMA2 (Touvron et al., 2023b)

4096. LLaMA3 series (Dubey et al., 2024) progressively increases the context length from 8K to 128K during pretraining. With pretrained models supporting longer context length, long context finetuning is playing an increased role in adapting pretrained LLMs; for example with datasets related to instruction finetuning and chain-of-thought. Furthermore, finetuning models are not bounded by their pretrained context lengths. Works have been done to adapt a pretrained LLM to a context length longer than that of the pretraining stage (Jin et al., 2024).

Transformer-based diffusion models also benefit from longer sequence lengths. DiT (Peebles & Xie, 2023) demonstrated that reducing the patch size, thereby increasing the sequence length of patches, leads to a significant improvement in image generation quality.

**Growth of activations.** With both batch size $B$ and sequence length $S$ being pushed up by the current trend, from formula 1 it is immediately clear that the size of activations would continue to grow over time.

## 2.3 ACTIVATIONS V.S. WEIGHTS

**Activation-to-weight ratio.** Unlike activations, the dimension $D$ of a model's weights remains fixed across different finetuning runs. This static nature of weights contrasts with the dynamic growth of activations, which scale with batch size $B$ and sequence length $S$.

Consider the size of the intermediate activations, which can be represented as $B \times S \times D$. For simplicity, suppose this corresponds to a linear layer of size $D \times D$ producing these activations. Then the ratio between the size of the activations and the size of its corresponding weights can be expressed as:

$$\frac{B \times S \times D}{D \times D} = \frac{B \times S}{D} \tag{2}$$

As illustrated in the previous section, batch size $B$ and sequence length $S$ would continue to be scaled up in the forseeable future. Thus for any existing model, future finetuning runs would possibly have a larger activation-to-weight ratio.

**Smaller models.** Smaller models have been designed to handle increasingly longer context lengths. For example, models like LLaMA-3.2-1B are capable of handling sequence lengths of 128K tokens, far longer than its larger predecessors. This trend demonstrates that improvements in model architecture and training techniques have allowed for smaller models (and thus smaller $D$) to manage longer contexts, further driving up the activation-to-weight ratio in formula 2.

## 2.4 OPERATING ON WEIGHTS INSTEAD OF ACTIVATIONS

**Activation-based paradigm.** Currently, PyTorch is based on activations. Weights are implemented as functions that operates on activations. The implementation paradigm of a neural network in PyTorch is to iteratively apply different operations to the intermediate activation $x$.

**Activation-based PEFT implementation.** The current PEFT implementation follows the activation-based paradigm during training. For example, in the standard LoRA implementation, LoRA operates at the level of the activations as it injects a low-rank update into the activation flow rather than directly manipulating the weight matrices. This LoRA implementation falls into the category where a reformulation to weights is possible, which will be illustrated in the next section.

**Weight-based operations.** During the forward pass during training, if an operation on activations could be reformulated to an operation on its corresponding weights, then the intermediate representation that needs to be stored shifts from an activation to the weight. The size of the intermediate representation would decrease when activations are larger than weights.

## 3 METHOD

### 3.1 LORA REFORMULATION

Consider a standard LoRA training setting given the pretrained weights $W \in \mathbb{R}^{D' \times D}$. The input tensor $x$ is a three-dimensional tensor $x$ of shape $(B, S, D)$, where $B$ is the batch size, $S$ is the

Figure 3: **Weight-based LoRA reformulation.** The left figure illustrates the weight-based reformulation of LoRA during training, while the right figure presents the activation-based reformulation.

sequence length, and $D$ is the hidden size. In our approach, instead of computing activations for the lora branch and the main path separately, we reformulate it into merging the product of $BA$ into $W$.

$$Wx + BAx \Rightarrow (W + BA)x$$

Note that this is different from original LoRA setting, where merging only happens for inference. As shown in Figure 3, in our formulation, $(W + BA)$ gets calculated first, and the multiplication with the input tensor only happens once, in contrast to twice ($Wx$ and $BAx$) in the original setting.

The key difference lies in the intermediate tensors created: the weight-based method primarily generates $BA$ and $W + BA$, while the activation-based approach produces $Wx$, $Ax$, $B(Ax)$. As batch size and sequence length increase, the dimensions of these activation tensors grow proportionally, leading to significantly higher memory consumption in the activation-based method compared to the weight-based approach, where the size of $W + BA$ remains constant regardless of input dimensions. This provides compelling motivation for the weight-based method's improved performance which we further analyze empirically.

### 3.2 REFORMULATION FOR OTHER METHODS

| Method | Original Formulation | Weight-based reformulation |
|---|---|---|
| *Format* | $y = Wx + b$ | $y = W'x + b'$ |
| LoRA[*] | $y = Wx + b + BAx$ | $y = (W + BA)x + b$ |
| (IA)$^{3}$[†] | $y = (Wx + b) \odot l$ | $y = (W \, diag(l))x + (b \odot l)$ |
| VeRA[◇] | $y = Wx + b + \Lambda_A A \Lambda_B Bx$ | $y = (W + \Lambda_A A \Lambda_B B)x + b$ |
| LoReFT[△] | $y = h + R_L^T(W_L h + b_L - R_L h)$ | $y = (R_L^T W_L W)x +$ |
| | $h = Wx + b$ | $(R_L^T W_L b + R_L^T b_L)$ |
| DiReFT[‡] | $y = h + W_{D,2}^T(W_{D,1}h + b_D)$ | $y = ((\mathbf{I} + W_2^T W_1)W)x +$ |
| | $h = Wx + b$ | $((\mathbf{I} + W_2^T W_1)b + W_2^T b_D)$ |

Table 1: **Weight-based reformulation.** Many PEFT techniques can be reformulated to be weight-based. [*](Hu et al., 2021), [†](Liu et al., 2022), [◇](Kopiczko et al., 2024), [△](Zhengxuan Wu, 2024), [‡](Zhengxuan Wu, 2024).

The weight-based reformulation principle we've introduced extends beyond LoRA-based techniques, encompassing a diverse range of Parameter-Efficient finetuning (PEFT) methods. Table 1 summarizes the original formulations and their corresponding weight-based reformulations for various PEFT techniques. Notably, this reformulation strategy applies not only to additive methods like LoRA (Hu et al., 2021) and VeRA (Kopiczko et al., 2024), but also to IA$^3$ (Liu et al., 2022), which was originally designed to scale activations element-wise. Our reformulation incorporates this scaling directly into the weight matrix using a diagonal matrix multiplication. Furthermore, we show that even more complex methods like LoReFT and DiReFT (Zhengxuan Wu, 2024), which weren't originally designed to be merged into the base model and instead manipulate activations of representation layers, can be reformulated in this way. Our weight-based reformulation encompasses a wide spectrum of PEFT techniques while maintaining their semantic equivalence.

In the following section, we present results comparing our weight-based reformulation to the original formulation for both IA³ and LoRA. These two PEFT techniques were chosen because of their popularity and compatiblity with Huggingface's PEFT library.

## 4 EXPERIMENTS

We conduct evaluations to understand the empirical effects of our reformulation on popular benchmarks and LLMs.

### 4.1 EXPERIMENTAL SETUP

**Language modeling.** To understand our weight-based reformulation's practical impact, we choose popular models paired with established datasets. Specifically, we employ the models LLaMA-3.2-1B and LLaMA-3.2-3B. These models have hidden dimensions of 2048 and 3072, respectively. For datasets, we use the MATH (Hendrycks et al., 2021), wikitext (Merity et al., 2016), and CNN-DailyMail (Nallapati et al., 2016) datasets. The maximum sequence lengths, in number of tokens, for these datasets vary per model tokenizer and are provided with the results. The tokenization of these datasets create sequences whose length closely mirror the sequence lengths of data that the model may encounter in practical scenarious.

**Diffusion models.** Diffusion finetuning using LoRA is actively supported in huggingface diffusers library (von Platen et al.). We study our reformulation on diffusion transformers (DiT) (Peebles & Xie, 2023). Transformer-based diffusion models has been shown to have good scaling performance on FLOPs, which relies heavily on sequence length. A smaller patch size directly translates to slicing images into more patches, resulting in a larger sequence length for diffusion transformers. We conduct experiments on DiT-S/2, DiT-B/2, DiT-L/2, DiT-XL/2, whose parameter counts range from 33M to 675M. The embedding dimensions range from 384 to 1152.

**Baselines and measurements.** We apply our reformulation to the PEFT methods $IA^3$ and LoRA ($r = 8$, applied to all linear layers). We compare our reformulation to the traditional weight-based approach as well as full-parameter finetuning. We use the Huggingface `peft` (Mangrulkar et al., 2022) library for our LLM activation-based PEFT implementations and a custom PEFT implementation for diffusion transformers. For memory measurements, we record the maximum memory usage across all CUDA GPUs. For latency measurements, we measure the time, in seconds, per batch, averaged over twenty batches.

### 4.2 DOES THE WEIGHT-BASED REFORMULATION SAVE MEMORY?

The memory savings of our weight-based reformulation of two PEFT techniques on LLMs are presented in Table 2, which shows the peak required GPU memory for a particular model, dataset, and batch size combination. This reformulation saves memory as compared to the activation-based baseline in every combination. As hypothesized, the savings of the weight-base reformulation over the activation-based approach increase proportionally as batch size and sequence length increase. Simultaneously, it also benefits from the smaller hidden dimension of the smaller models. For example, when finetuning LLaMA-3.2-3B on the MATH dataset with a batch size of 16, **the activation-based approach uses 1.4x more memory than our reformulation.** These savings are important; with them, one could elect to increase their batch size, train on longer sequences, or even opt for a larger model.

Furthermore, with this reformulation, LoRA and $IA^3$ now match or even beat the memory usage of full finetuning. This is to be expected: the memory that would be used by the Adam optimizer (2x the size of the model weights) for finetuning all parameters is compensated for by the introduction of the intermediate tensors $BA$ and $(W + BA)$ on the computation graph, each of which is 1x the size of the model weights.

We seek to see if these results hold for diffusion transformers. As shown in Table 3, across DiT variants of different sizes, there is consistent memory savings by our weight-based reformulation. For these models, We see that the activation-based approach consumes, at minimum, 1.24x more memory for $IA^3$ and 1.13x more memory for LoRA than our weight-based reformulation for diffusion

| Batch Size | PeFT | LLaMA-3.2-1B | | | LLaMA-3.2-3B | | |
|---|---|---|---|---|---|---|---|
| | | wikitext | MATH | CNN/DM | wikitext | MATH | CNN/DM |
| **Maximum Sequence Length** | | 865 | 2566 | 4185 | 865 | 2566 | 4185 |
| 8 | full fine-tuning | 24.9 | 68.2 | 110.8 | 42.2 | 114.5 | 185.5 |
| | $IA^3$ (original) | 27.9 | 79.8 | 131.7 | 49.0 | 136.9 | 228.0 |
| | $IA^3$ (ours) | 25.3 | 67.3 | 108.1 | 45.1 | 112.8 | 179.4 |
| | Memory Efficiency (x) | **1.10** | **1.19** | **1.22** | **1.09** | **1.21** | **1.27** |
| | LoRA (original) | 29.3 | 83.5 | 136.2 | 54.3 | 151.2 | 237.2 |
| | LoRA (ours) | 25.4 | 67.3 | 110.2 | 45.2 | 112.9 | 179.5 |
| | Memory Efficiency (x) | **1.15** | **1.24** | **1.26** | **1.20** | **1.34** | **1.32** |
| 16 | full fine-tuning | 46.5 | 157.5 | - | 78.4 | 226.0 | - |
| | $IA^3$ (original) | 53.5 | 188.5 | - | 91.9 | 278.3 | - |
| | $IA^3$ (ours) | 46.4 | 153.6 | - | 79.0 | 217.4 | - |
| | Memory Efficiency (x) | **1.15** | **1.23** | - | **1.16** | **1.28** | - |
| | LoRA (original) | 56.5 | 189.9 | - | 102.3 | 303.9 | - |
| | LoRA (ours) | 46.5 | 153.7 | - | 79.1 | 217.5 | - |
| | Memory Efficiency (x) | **1.22** | **1.24** | - | **1.29** | **1.40** | - |

Table 2: We record the maximum memory usage (GiB) of running finetuning on a given batch size, model, and dataset combination. We do this for two techniques, $IA^3$ and LoRA. We then calculate how much worse the original formulation performs as a multiple of our weight-based-formulation's performance in the *Memory Efficiency (x)* rows. Out of memory errors are denoted with a dash.

| PeFT Method | DiT-S/2 (33M) | DiT-B/2 (130M) | DiT-L/2 (458M) | DiT-XL/2 (675M) |
|---|---|---|---|---|
| full fine-tuning | 4.4 | 9.6 | 26.6 | 36.0 |
| LoRA (original) | 4.6 | 9.6 | 25.8 | 34.3 |
| LoRA (ours) | 4.1 | 8.5 | 22.8 | 30.3 |
| Memory Efficiency (x) | **1.14** | **1.14** | **1.13** | **1.13** |
| $IA^3$ (original) | 5.1 | 10.5 | 28.1 | 37.3 |
| $IA^3$ (ours) | 4.0 | 8.5 | 22.7 | 30.2 |
| Memory Efficiency (x) | **1.26** | **1.25** | **1.24** | **1.24** |

Table 3: Maximum memory usage (GiB) of Diffusion Transformers finetuned with a fixed batch size of 1. The input resolution given to all models is of the same size $256 \times 256$.

models. Additionally, whereas the activation-based approach for $IA^3$ was more memory expensive than full-finetuning, with the weight-based reformulation it is now more memory-efficient.

Finally, we noticed that, empirically, these results have some deviation. This behavior is primarily due to the CUDA backend of PyTorch, whose kernels have been optimized to speed up computation at the expense of allocating additional memory in certain programs. While out of scope for this work, these features explain marginal additional memory allocations.

### 4.3 DOES THE WEIGHT-BASED REFORMULATION IMPROVE LATENCY?

Transferring data between the global memory and the tensor cores can induce a significant latency overhead. Additionally, smaller tensors mean fewer FLOPs during matrix multiplication. Given that our approach is more memory efficient, we hypothesize that, for this reason, it could induce lower latency as well.

Table 4 shows the second per batch for our weight-based reformulation versus the activation-based approach on LoRA and $IA^3$. The results in this table empirically confirm this hypothesis, showing that our reformulation is consistently faster than the activation-based approach. Specifically, for finetuning LLaMA-3.2-1B on wikitext, **the activation-based approach can take as much as 1.4x longer to complete a batch than our weight-based reformulation.** This conclusion is further compounded by the latency improvements on diffusion models as presented in Table 5, where similarly the activation-based approach can take as much as 1.39x longer to complete a batch than

| Batch Size | PeFT | LLaMA-3.2-1B | | | LLaMA-3.2-3B | | |
|---|---|---|---|---|---|---|---|
| | | wikitext | MATH | CNN/DM | wikitext | MATH | CNN/DM |
| **Maximum Sequence Length** | | 865 | 2566 | 4185 | 865 | 2566 | 4185 |
| 8 | full fine-tuning | 0.73 | 2.17 | 3.51 | 1.71 | 5.74 | 9.27 |
| | $IA^3$ (original) | 0.73 | 2.56 | 3.89 | 1.52 | 5.24 | 8.05 |
| | $IA^3$ (ours) | 0.65 | 2.06 | 3.30 | 1.42 | 4.71 | 7.40 |
| | Latency Speedup (x) | **1.13** | **1.24** | **1.18** | **1.07** | **1.11** | **1.09** |
| | LoRA (original) | 0.89 | 3.12 | 4.48 | 1.89 | 6.20 | 9.22 |
| | LoRA (ours) | 0.68 | 2.44 | 3.57 | 1.49 | 4.82 | 7.69 |
| | Latency Speedup (x) | **1.30** | **1.28** | **1.26** | **1.33** | **1.29** | **1.20** |
| 16 | full fine-tuning | 1.26 | 5.12 | - | 2.86 | 9.90 | - |
| | $IA^3$ (original) | 1.31 | 5.15 | - | 2.84 | 9.83 | - |
| | $IA^3$ (ours) | 1.16 | 4.48 | - | 2.60 | 9.28 | - |
| | Latency Speedup (x) | **1.13** | **1.15** | - | **1.09** | **1.06** | - |
| | LoRA (original) | 1.58 | 6.00 | - | 3.76 | 11.69 | - |
| | LoRA (ours) | 1.19 | 4.65 | - | 2.70 | 9.18 | - |
| | Latency Speedup (x) | **1.33** | **1.29** | - | **1.40** | **1.27** | - |

Table 4: We record the latency in terms of seconds per batch of running LoRA and $IA^3$ finetuning on a given batch size, model, and dataset combination on. We then calculate how much worse the original formulation performs as a multiple of our weight-based-formulation's performance in the *Latency Speedup (x)* rows. Out of memory errors are denoted with a dash.

| PeFT Method | DiT-S/2 (33M) | DiT-B/2 (130M) | DiT-L/2 (458M) | DiT-XL/2 (675M) |
|---|---|---|---|---|
| full fine-tuning | 0.66 | 0.99 | 1.15 | 1.94 |
| LoRA (original) | 0.31 | 0.73 | 1.64 | 2.26 |
| LoRA (ours) | 0.28 | 0.56 | 1.18 | 1.79 |
| Latency Speedup (x) | **1.12** | **1.29** | **1.39** | **1.25** |
| $IA^3$ (original) | 0.36 | 0.73 | 1.26 | 1.86 |
| $IA^3$ (ours) | 0.36 | 0.73 | 1.34 | 1.90 |
| Latency Speedup (x) | **1.06** | **1.17** | **1.07** | **1.02** |

Table 5: We record the latency in terms of seconds per batch of running LoRA and $IA^3$ finetuning on a given model. We calculate how much worse the original formulation performs as a multiple of our weight-based-formulation's performance in the *Latency Speedup (x)* rows. Out of memory errors are denoted with a dash.

the weight-based reformulation, particularly on DiT-L/2. Given that access to training hardware can be limited, this impact is important for minimizing training time.

## 5 ANALYSIS

To better characterize the nature of the relationship between the weight-based reformulation versus the activation-based approach, we conduct ablation studies across rank, batch size, and sequence length.

### 5.1 LORA RANK

Figure 4 shows the changes in memory consumption in our weight-based reformulation versus the activation-based approach as $r$ is varied. The results in this table show that the rank of LoRA matrices does not contribute significantly to the overall memory consumption for values of $r$ less than or equal to 128. We also observed that as the rank increases, the original implementation increases more rapidly than our reformulation. This trend is due to the intermediate tensor generated by the $Ax$ multiplication in $Wx + BAx$, in which one of the dimensions is of size $r$ in the activation based formulation of LoRA. We note that both methods see in increase in memory consumption as $r$

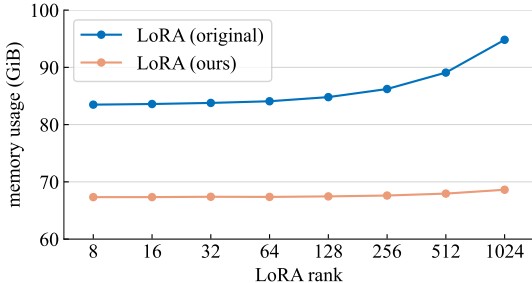

Figure 4: **LoRA rank.** The weight-based reformulation prevents the need to save the activation produced after the down projection $Ax$. Thus, when using higher LoRA rank, our reformulation saves additional memory.

increases due to the optimizer states accounting for more trainable parameters. From this experiment it is clear that the weight-based formulation is not significantly dependent on the rank used by LoRA, offering an avenue to explore higher values of $r$ without trading off memory.

## 5.2 BATCH-SIZE

| Batch Size | 1 | 2 | 4 | 8 | 16 | 32 |
|---|---|---|---|---|---|---|
| LoRA (ours) | 19.0 | 20.5 | 23.4 | 29.4 | 54.5 | 104.9 |
| LoRA (original) | 19.0 | 20.3 | 24.9 | 34.1 | 65.6 | 128.8 |
| Memory Savings (x) | **1.00** | **0.99** | **1.06** | **1.16** | **1.20** | **1.23** |

Table 6: Maximum memory usage (GiB) for training LLaMA-3.2-1B with varying batch sizes.

Table 6 illustrates the memory usage as we vary the batch size when training on LLaMA 3.2 1B on wikitext, comparing our LoRA reformulation to the original method. At smaller batch sizes, the memory savings from our approach are modest, nearly identical to the original. However, as the batch size increases, our weight-based reformulation demonstrates significant improvements in memory efficiency. These trends suggest that our reformulation is particularly well-suited for larger batch sizes which is crucial for faster and more stable training.

## 5.3 SEQUENCE LENGTH

| Sequence Length | 256 | 512 | 1024 | 2048 | 4096 |
|---|---|---|---|---|---|
| LoRA (original) | 11.79 | 21.13 | 39.83 | 77.22 | 152 |
| LoRA (ours) | 11.69 | 19.05 | 33.97 | 63.8 | 123.46 |
| Memory Savings (x) | **1.01** | **1.11** | **1.17** | **1.21** | **1.23** |

Table 7: Maximum memory usage (GiB) for training LLaMA-3.2-1B with varying sequence lengths.

Table 7 demonstrates how memory is affected as we increase the sequence length of the data that we finetune on using our weight-based reformulation versus the activation-based approach. We created a fake dataset with randomly sampled tokens of a specified sequence length. The trend supports the hypothesis: as we increase the sequence length, we see increasing gains from our reformulation in memory savings. We only go up to a sequence length of 4096, but LLaMA-3.2-1B's maximum sequence length is 128,000. As inputs to these models trend larger, particularly in the age of chain-of-thought reasoning and agents, being able to effectively manage the memory for finetuning on these longer sequences becomes even more important.

## 6   RELATED WORKS

**PEFT + Compression.**  Methods such as QLoRA Dettmers et al. (2024) and LoftQ (Li et al., 2024) reduce model size in memory through low-bit quantization of model parameters, while reducing information with LoRA. Similarly, Wanda (Sun et al., 2024) uses pruning to reduce the number of active weights in the model and uses LoRA to regain lost model performance. While these techniques reduce memory footprints via model compression, our approach further complements these by decreasing the memory consumption during post-compression training. Our contribution furthers the memory efficiency of finetuning with compression without sacrificing performance.

**Finetuning Frameworks.**  Finetuning frameworks like DeepSpeed (Rasley et al., 2020) and ZeRO (Rajbhandari et al., 2020) focus on infrastructure-level optimizations to improve parallelism and latency efficiency by sharding optimizer states across accelerators. Our method is designed to integrate with these frameworks. These frameworks take a fixed computation graph and optimize the training system over this graph. Our technique aims to build on top of these methods by providing a more efficient computation graph to the frameworks.

**Compilers.**  Other methods have approached efficient training through compilation. PyTorch2 (Ansel et al., 2024) introduced TorchDynamo, a just-in-time compiler for computation graphs. While these works focus on speeding up training time, our work complements it by addressing memory consumption. As a byproduct of our work, we also improve on latency, as shown in 4.3.

## 7   LIMITATIONS AND CONCLUSION

We acknowledge that our work is not without its limitations. It is important to recognize that the benefits of our reformulation can diminish, particularly in situations where the batch size and sequence length is small or when the hidden dimension size is large. We believe the field is moving away from this paradigm as smaller models become more capable and contexts become longer. Additionally, although not being actively used, the current dropout implementation in LoRA is only applied to activations that pass through the LoRA branch, while not being applied to the input to the frozen weight. Therefore, the activation-based LoRA dropout implementation would make weight-based reformulation not exactly equivalent to the LoRA when using dropout. Nonetheless, Lin et al. (2024) proposes to apply dropout to LoRA weights, which is compatible with our weight-based reformulation.

Parameter-efficient finetuning techniques have prevailed for their ability to reduce memory usage during finetuning as well as overfitting (Biderman et al., 2024). In this study, we offer an equivalent reformulation for these these techniques that further reduces their memory usage and latency. Our main insight is the trend that capable, state-of-the-art models are getting smaller while their context lengths are getting larger. This insight means that activations are trending towards becoming larger than the size of the weights of the model. We leverage this information to have these PEFT techniques not produce activations, but rather a delta to the weights they are being applied to. Our experiments demonstrate that this leads to memory and latency savings as sequence lengths and batch sizes increase. We hope that that this research provides greater accessibility to finetuning and inspires future research in optimizing PEFT methods.

**Reproducibility Statement:** Our code could be found in the anonymous GitHub repository: https://anonymous.4open.science/r/slimscale-19B6. It is also included in the supplementary material.

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

## A  APPENDIX

### A.1  FIGURE 1 SET UP

In figure 1, the plot on the left illustrates the maximum memory usage during fine-tuning on a synthetic dataset with a sequence length of 512 and a batch size of 1, across LLaMA models of varying sizes: 8B, 3B, and 1B. The right plot displays the maximum GPU memory usage as context length varies, also on a synthetic dataset, for the LLaMA 3.2 1B model with a batch size of 1.

## A.2 SINGLE-LAYER ANALYSIS

**Settings.** We study the memory usage when applying LoRA finetuning on a linear layer with $W \in \mathbb{R}^{2048 \times 2048}$, which is of the same dimension as the query projection layer $W_q$ in LLaMA-3.2-1B. We also used a LoRA rank of 8, batch size of 32, and sequence length of 1024. This setup allowed us to isolate the memory behaviors of the two LoRA variants while running on a CPU to avoid GPU-specific optimizations. We use the PyTorch profiler to track peak memory usage during forward and backward passes, and the script can be found in our anonymous repo in a file called *single_layer_analysis.py*.

**Memory Profile.** In the weight-based formulation, the extra operations primarily involve matrix multiplication BA and addition $W + BA$, which each consume 16 Mb. Additionally, the linear transformation $(W + AB)x$ takes 64 Mb of memory. In contrast, the activation-based formulation introduces additional activations of much larger size, including $Wx$ of size 64 mb, and $B(Ax)$ and the subsequent addition $Wx + B(Ax)$, each of size 128 Mb.

There is a significant difference in peak memory consumption between the two formulations: the weight-based approach reaches a maximum of 160.125 Mb, while the activation-based method peaks at 256.63 Mb of CPU memory. This 60% increase in memory usage for the activation-based approach can be primarily attributed to the storage requirements for intermediate activations such as Wx and B(Ax).

It is worth noting, however, that the practical implementation of these approaches may be influenced by CUDA optimizations and PyTorch optimizations, which can introduce additional nuances to the memory usage patterns. Nonetheless, this simplified single-layer analysis motivates the potential benefits of the weight-based LoRA formulation.