# OpenReview forum: "Activations Aren't Cheap in LoRA, Weights Are"
_ICLR.cc/2025/Conference — ICLR 2025 Conference Withdrawn Submission_

### Official Review · Reviewer_fFSm · 2024-10-16

**Soundness:** 1
**Presentation:** 2
**Contribution:** 2
**Rating:** 5
**Confidence:** 3

**Summary:**

This paper proposes a reformulation method to improve the LoRA and some PEFT techniques used in LLM finetuning. It reformulates the activation-based method to weight-based to reduce memory consuming and latency for longer sequence length and larger batch size.

**Strengths:**

The finetuning of LLMs for long sequence length is an important problem, and the proposed method is clear and reasonable.

**Weaknesses:**

1. The proposed reformulation is designed based on long sequence length situations, but the maximum sequence length of the datasets used in evaluation is even less than 8192. Datasets with longer sequence length are necessary to verify the clarification. Can you explain why the trends observed in that study would extend to even longer sequences like 128K?

2. The estimation of activation-to-weight ratio is not fair enough. Computing of attention scores have a different pattern with linear layers, and results in quiet different ratio in LLMs. Can you provide a breakdown of the activation-to-weight ratios for different components of the model, including attention layers and feed-forward layers?

3. The configuration of experiments is not clear. Which type and how many GPUs are used? Any distibuted training methods used? These configurations can significantly affect the performance. Can you give more experiments setup details and outline the hardware configuration, distributed training setup (if any), and any other relevant implementation details that could affect performance?

4. From the evaluation results, the proposed reformulation does improve the original method but show little advantage over full fine-tuning on both memory and latency. Can you provide a more comprehensive comparison between your method and full fine-tuning, including accuracy metrics? Can you justify why your improved methods are better than full fine-tuning under different configuration of sequence length, batch size, and model size?

5. Figure 1 lacks information. What’s the sequence length for left part and what’s the model size for right part? The explanation should have been in the caption instead of appendix. It’s also confusing that the markers in two parts are different.

6. Caption of Figure 3 mismatches the figure. Should be "top and bottom" instead of "left and right".

**Questions:**

1. Can you clarify why the current conclusion can be extended to even longer sequence length like 128K? The current setup of experiments and analysis are not convicing enough to me. It will help a lot if you can offer evaluation results on longer sequence length and provide a breakdown of the "activation-to-weight ratios" in different components of LLMs.

2. Can you justify why your method is better than full fine-tuning? Again, how the conclusion can be extended to longer sequence length?

---

### Official Review · Reviewer_dX6w · 2024-11-03

**Soundness:** 3
**Presentation:** 2
**Contribution:** 2
**Rating:** 3
**Confidence:** 4

**Summary:**

This paper highlights the increasing demand for memory as context lengths expand in transformers, which could impact fine-tuning on consumer GPUs. While LoRA has provided memory savings, its benefits may diminish with trends toward smaller models and longer contexts. The proposed weight-based reformulation, which merges the LoRA branch with the main path by combining BA into W, effectively reduces memory usage and latency across various Parameter-Efficient Fine-Tuning (PEFT) methods. Experiments show that this approach uses significantly less memory and time compared to the activation-based approach, as demonstrated on tasks such as language modeling, diffusion models, and using LLaMA-3.2 models.

**Strengths:**

1. The simple idea improves the efficiency of activation memory in traditional PEFT methods (such as LoRA) and reduces running latency.

2. The proposed idea can be easily applied to extended PEFT methods of LoRA, such as IA, VeRA, and LoReFT.

3. Sharing the code implementing this idea enhances the credibility of the evaluation results.

**Weaknesses:**

1. The activation-based and weight-based paradigms should be adopted depending on different scenarios. For example, the activation-based paradigm is useful when the activation size is large compared to the model size, but it becomes disadvantageous in the opposite case. Although the authors qualitatively mentioned this, they did not provide a quantitative analysis or experiments to determine in what range the proposed activation-based paradigm would be beneficial, which reduces the algorithm's practical utility in real-world applications.

2. It is also questionable whether scenarios involving small models with large activations (such as large batch sizes or long sequence lengths) occur frequently enough to justify the activation-based paradigm's effectiveness. Generally, small LLM models are designed for edge devices, where limited memory makes it challenging to support long sequences. In such cases, the scenario would likely involve a small model with small activations, raising doubts about the activation-based paradigm's effectiveness in these instances.

3. Upon reviewing the provided code, it appears challenging to apply dropout to the activation results generated after the weight-based reformulation. This may impact the fine-tuning results, suggesting a lack of evidence for the claim that the proposed method does not affect model performance, as no accuracy comparison experiments were conducted.

**Questions:**

1-3. The primary concern is related to the weaknesses above. If these concerns are adequately answered, I am willing to consider increasing the score.

4. Could you provide a detailed comparison of activation memory and weight memory, in addition to the maximum memory usage shown in Table 2 and Table 3?

5. It would be beneficial to mention methods like gradient checkpointing [1] and activation compressed training [2-6] in the related works section. Additionally, experimental results comparing these methods would strengthen the work. If direct one-to-one comparison is challenging due to differences in scope, please clarify that these methods are orthogonal and provide experimental results that apply these techniques in combination with the proposed method. If time is limited, it would be helpful to demonstrate that the activation-based paradigm is more efficient than the weight-based paradigm when gradient checkpointing [1] and GACT [3] are applied.

[1] Training Deep Nets with Sublinear Memory Cost, arxiv, 2016.
[2] ActNN: Reducing Training Memory Footprint via 2-Bit Activation Compressed Training, ICML, 2021.
[3] GACT: Activation Compressed Training for Generic Network Architectures. ICML, 2022
[4] Learning with Auxiliary Activation for Memory-Efficient Training, ICLR, 2023.
[5] DropIT: Dropping Intermediate Tensors for Memory-Efficient DNN Training, ICLR, 2023.
[6] ALAM: Averaged Low-Precision Activation for Memory-Efficient Training of Transformer Models, ICLR, 2024

---

### Official Review · Reviewer_L3WB · 2024-11-04

**Soundness:** 2
**Presentation:** 3
**Contribution:** 2
**Rating:** 3
**Confidence:** 3

**Summary:**

The paper "Activations Aren't Cheap in LoRA, Weights Are" presents a method to address the issue of high memory consumption associated with activations in smaller large language models (LLMs) with extended context lengths. The authors propose a reformulation that focuses on manipulating model weights instead of activations during fine-tuning, aiming to reduce the memory overhead that grows with increasing context lengths. This weight-based approach is designed to offer memory savings and improved latency, particularly in scenarios where memory resources are constrained.

**Strengths:**

1) Improved Latency and Performance: The weight-based method reduces memory consumption and latency. This improvement is particularly valuable for users aiming to fine-tune models on consumer hardware with limited resources.

2) Applicability to Other Fine-Tuning Methods: The reformulation is applicable to other parameter-efficient fine-tuning (PEFT) methods, not just LoRA.

**Weaknesses:**

1) Limited Novelty: The approach is primarily a reformulation, shifting operations from activations to weights. While practical, it doesn’t introduce new insights or innovative techniques.

2) Limited Impact: The benefits of this work are mainly applied to smaller models with long context lengths on memory-constrained hardware, so the impact is somewhat narrow and may not generalize to larger models or different settings.

**Questions:**

1) Does this reformulation introduce any risks of numerical instability?

---

### Official Review · Reviewer_BKEb · 2024-11-06

**Soundness:** 3
**Presentation:** 3
**Contribution:** 2
**Rating:** 6
**Confidence:** 4

**Summary:**

The paper proposes a weight-based reformulation of LoRA to reduce the memory overhead of LoRA fine-tuning, which can be worse than that of full model fine-tuning in certain scenarios. The proposed reformulation is mathematically equivalent (assuming no dropout), and thus can offer both memory and latency savings for free.

**Strengths:**

- The paper tackles an important topic: improving the memory efficiency of fine-tuning LLMs/LVMs
- The proposed technique is simple and easy to understand
- The paper is well-written
- The experiments show that the proposed method does improve both latency and memory of LoRA fine-tuning

**Weaknesses:**

- The paper studies the extreme case of adding LoRA to all linear layers. In practice, LoRA layers are added to a select layers (typically attention layers only). How would the memory profiling look like under that setting? Do the current experiments also add LoRA to the final output embedding matrix?
- The method only works when no extra transformations are applied to the LoRA hidden representation (e.g., dropout).
- The datasets being used in Tables 2 and 4 are pointless, what is shown is profiling of a single batch, this could be done with randomized data.

**Questions:**

- What was the optimizer used for these experiments? I suspect this is not Adam, since the memory overhead of the optimizer states would heavily impact the memory requirements of full model fine-tuning. Can the authors reproduce results of full model fine-tuning, LoRa (both version) with the Adam optimizer?

---

### Author Response · Authors · 2024-11-20
**Withdrawing Submission**

We sincerely thank the reviewers and the area chair for reviewing and meta-reviewing our work. We greatly appreciate the time and effort you dedicated to providing us with valuable feedback.

We have realized that the comparison involving LoRA in our work is not entirely fair due to a mismatch between HuggingFace’s implementation and our own, specifically in how the data types of the LoRA adapters are initialized. As a result, the LoRA results in the main table do not accurately reflect a fair comparison. While the IA$^3$ results remain valid, we have decided to withdraw the paper.

We thank you all again for your time and efforts.

---

### Note · Authors · 2024-11-20

I have read and agree with the venue's withdrawal policy on behalf of myself and my co-authors.